# Cognitive Training as a Potential Activator of Hippocampal Neurogenesis in the Rat Model of Sporadic Alzheimer’s Disease

**DOI:** 10.3390/ijms21196986

**Published:** 2020-09-23

**Authors:** Alena O. Burnyasheva, Tatiana A. Kozlova, Natalia A. Stefanova, Nataliya G. Kolosova, Ekaterina A. Rudnitskaya

**Affiliations:** Institute of Cytology and Genetics, Siberian Branch of Russian Academy of Sciences (ICG SB RAS), 10 Lavrentyeva Ave., 630090 Novosibirsk, Russia; burnyasheva@bionet.nsc.ru (A.O.B.); kozlova@bionet.nsc.ru (T.A.K.); stefanovan@bionet.nsc.ru (N.A.S.); kolosova@bionet.nsc.ru (N.G.K.)

**Keywords:** cognitive training, neurogenesis, dentate gyrus, amyloid-β, Alzheimer’s disease, OXYS rats

## Abstract

There is a growing body of evidence that interventions like cognitive training or exercises prior to the manifestation of Alzheimer’s disease (AD) symptoms may decelerate cognitive decline. Nonetheless, evidence of prevention or a delay of dementia is still insufficient. Using OXYS rats as a suitable model of sporadic AD and Wistar rats as a control, we examined effects of cognitive training in the Morris water maze on neurogenesis in the dentate gyrus in presymptomatic (young rats) and symptomatic (adult rats) periods of development of AD signs. Four weeks after the cognitive training, we immunohistochemically estimated densities of quiescent and amplifying neuronal progenitors, neuronal-lineage cells (neuroblasts and immature and mature neurons), and astrocytes in young and adult rats, and the amyloid precursor protein and amyloid-β in adult rats. Reference memory was defective in OXYS rats. The cognitive training did not affect neuronal-lineage cells’ density in either rat strain either at the young or adult age, but activated neuronal progenitors in young rats and increased astrocyte density and downregulated amyloid-β in adult OXYS rats. Thus, to activate adult neurogenesis, cognitive training should be started before first neurodegenerative changes, whereas cognitive training accompanying amyloid-β accumulation affects only astrocytic support.

## 1. Introduction

Alzheimer’s disease (AD) is a detrimental multifactorial disorder developing asymptomatically for many years prior to its manifestation [1]. There is no effective cure for AD at present; however, several individually modifiable interventions prior to manifestation of the disease symptoms are believed to improve cognitive function and to decrease the risk of AD [2,3,4]. Indeed, several studies have revealed positive associations of physical activity, cognitive training, or both with cognition in elderly people and patients with mild cognitive impairment [5,6,7]. These cognitive improvements may be explained by the activation of neuroplasticity. Currently, it is well known that adult neurogenesis occurring in the hippocampal dentate gyrus (DG) of mammals results in the integration of newborn granule cells into the hippocampus circuitry, thus providing an extra degree of plasticity that is crucial for the acquisition of certain types of contextual memory [8]. Physical exercise and cognitive training activate neurogenesis in the hippocampal DG of adult animals [9,10]. Nevertheless, it is still unclear at what age cognitive training should be started to improve cognitive function [11] because the precise mechanisms underlying cognitive improvement in elderly people are still unknown, and this research is difficult because of limitations of human studies as well as the absence of suitable animal models for the late-onset sporadic form of AD.

Previously, we have shown that senescence-accelerated OXYS rats may be considered an adequate model of late-onset sporadic AD because the disease signs in these rats develop spontaneously without mutations in genes *App*, *Psen1*, and *Psen2*. First neurodegenerative changes occur in OXYS rats at 3 months of age [12]; however, more recently, we have shown that neurodegeneration in OXYS rats is preceded by altered postnatal neurogenesis and a delay of hippocampal development. Indeed, we have shown increased density of neuroblasts and immature neurons at 10 days of age and an increased number of apoptotic cells at 20 days of age in the DG of OXYS rats as well as changes in neuronal-progenitor density and a delay of mossy-fiber formation [13,14]. Nevertheless, by the age of 1.5 months, differences between OXYS and control Wistar rats disappear [14]. Active accumulation of a toxic form of amyloid-β in the brain of OXYS rats is observed at 12 months of age and progresses until 18 months of age [15] against the backdrop of changes in the expression of genes associated with adult neurogenesis (according to a mammalian-adult-neurogenesis gene ontology database [16]) in the hippocampus [13]. Accordingly, in the present work, to assess the effects of cognitive training on adult hippocampal neurogenesis as one of neuroplasticity hallmarks, we used OXYS rats trained in the Morris ware maze (MWM) prior to development of neurodegeneration (1.5 months of age) and during active amyloid-β accumulation in the brain (12 months of age). MWM performance has been linked to long-term potentiation and NMDA receptor function, making it a key technique in the investigation of hippocampal circuitry [17,18].

## 2. Results

### 2.1. Spatial Learning, Reversal Learning, and Reference Memory of OXYS and Wistar Rats in the MWM

OXYS rats demonstrated some alterations of spatial learning in the MWM already at 1.5 months of age. Indeed, Wistar rats learned the location of the submerged platform starting from the second training day (*p* < 0.009); whereas OXYS rats started successfully finding the platform on the third training day (*p* < 0.02; Figure 1A). A move of the platform expectedly resulted in an increase of averaged latencies of platform finding for both rat strains: A 3.9-fold increase for Wistar rats (*p* < 0.004) and a 1.9-fold increase for OXYS rats (*p* < 0.02). By contrast, already on the seventh training day Wistar rats learned the new position of the platform (*p* < 0.05), whereas OXYS rats learned it on the eighth training day (*p* < 0.009).

On the last, eleventh, training day, the platform was removed from the MWM, and we analyzed the animals’ reference memory in the probe trial. We revealed successful reversal learning and reference memory in Wistar rats (Figure 1B): The animals preferred the platform area to other quadrants (*p* < 0.04 for all other quadrants). On the other hand, OXYS rats did not manifest successful reversal learning because of spending a larger amount of time in the quadrant where the platform was before the move (first quadrant) instead of the quadrant to which the platform was moved (third quadrant). Thus, we demonstrated a delay of learning in the MWM and altered reference memory in OXYS rats already at 1.5 months of age.

As for training in the MWM starting at 12 months of age, we detected no differences in spatial learning between OXYS and Wistar rats (Figure 1C): Animals of both strains learned the location of the submerged platform already starting on the second training day. After the move of the platform on the sixth training day, the averaged finding latency increased only in OXYS rats (*p* < 0.003). Nonetheless, already on the seventh training day, the parameter decreased to the level of Wistar rats and did not differ between rat strains until the end of the training.

In the probe trial (Figure 1D), Wistar rats spent twice as much time in the platform area (third quadrant) compared to the other quadrants (*p* < 0.05), whereas the time spent by OXYS rats in the third quadrant did not significantly differ from that spent in the first quadrant, which was the platform area before the platform move. These data may indicate that reference memory changed to some extent in OXYS rats at 12 months of age.

To summarize our results, Wistar rats demonstrated similar MWM performances at 1.5 and 12 months of age, with the exception of the absence of an averaged finding latency increase on the sixth training day after the move of the platform at 12 months of age. OXYS rats featured mild alterations of spatial learning and spatial reversal learning, as well as altered reference memory already at 1.5 months of age, in the period prior to neurodegeneration. At 12 months of age—the period of active amyloid-β accumulation in the brain—spatial learning and reversal learning abilities of OXYS rats did not differ from those of Wistar rats, but the reference memory of OXYS rats was found to be altered, as was the case at 1.5 months of age.

### 2.2. Effects of MWM Training on Cell Density in the Hippocampal DG of OXYS and Wistar Rats

The second stage of our work was aimed at the evaluation of the influence of MWM training on neurogenesis in the hippocampal DG via determination of densities of progenitors and immature and mature neurons and astrocytes at 4 weeks after the beginning of the MWM training. Therefore, the age of the rats advanced to 2.5 and 13 months; nontrained age-matched animals served as a control.

### 2.3. Densities of Neural Progenitors in the DG of OXYS and Wistar Rats

First, we analyzed density of quiescent neural progenitors (QNPs), which may develop into a neuronal or glial cell lineage. ANOVA revealed that the density of QNPs was affected by age (F_1.64_ = 36.7, *p* < 0.0001), genotype (rat strain; F_1.64_ = 12.6, *p* < 0.0007), and training (F_1.64_ = 5.8, *p* < 0.02), and there were interactions between genotype and age (F_1.64_ = 5.6, *p* < 0.02) as well as genotype and training (F_1.64_ = 5.8, *p* < 0.02). We did not find significant age-related changes in QNP density in the DG of Wistar rats; moreover, the MWM training did not affect QNP density in Wistar rats at any tested age; this finding may point to a stable QNP pool in the hippocampal neurogenic niche. By contrast, this was not the case for OXYS rats: Indeed, QNP density was higher than that in Wistar rats at 2.5 months of age (*p* < 0.0002; Figure 2A), and then the parameter decreased two-fold by 13 months of age (*p* < 0.0001; Figure 2A). As for the MWM training, it led to a decrease of QNP density in OXYS rats at 2.5 months of age (*p* < 0.004), likely indicating QNP activation.

Then, we estimated the density of amplifying neural progenitors (ANPs), which give rise to the neuronal cell lineage. We showed that ANP density decreased with age (F_1.64_ = 51.0, *p* < 0.0001); besides, there was an interaction between genotype and training (F_1.64_ = 7.8, *p* < 0.007). Pairwise comparisons uncovered a decrease in ANP density from 2.5 to 13 months of age in both rat strains (*p* < 0.0001 for Wistar rats and *p* < 0.01 for OXYS rats), indicating age-related slowdown of hippocampal neurogenesis. Nonetheless, in OXYS rats, this decrease was less pronounced because of lower ANP density at 2.5 months of age as compared to Wistar rats (*p* < 0.004; Figure 2A). The MWM training had opposite effects on ANP density between young OXYS and Wistar rats: Indeed, the training in the MWM caused a decrease of ANP density in Wistar rats (*p* < 0.0009) and an increase of this parameter in OXYS rats (*p* < 0.05). We did not observe any impact of the MWM training on ANP density in the DG of OXYS and Wistar rats at age 13 months (Figure 2A).

Taken together, these results may point to the enhancement of ANP differentiation into the neuronal cell lineage under the influence of the MWM training in young Wistar rats, with QNP density remaining unchanged. At the same time, in OXYS rats, the MWM training that was started at 1.5 months of age led to the activation of QNPs, thereby causing an increase of ANP density.

### 2.4. Densities of Neuroblasts and Immature and Mature Neurons in the DG of OXYS and Wistar Rats

Next, we analyzed the density of cells of the neuronal lineage: Neuroblasts, immature neurons, and mature neurons in the DG of OXYS and Wistar rats.

Surprisingly, factorial analysis indicated that the genotype, age, and learning did not affect the density of cells of the neuronal lineage; however, densities of neuroblasts decreased with age (F_1.38_ = 18.9, *p* < 0.0001; Figure 3A), which may reflect attenuation of hippocampal neurogenesis. As for the density of immature neurons, this parameter decreased 1.5-fold from 2.5 to 13 months of age in Wistar rats (insignificantly), whereas we did not observe any age-related changes in OXYS rats; thus, the parameter turned out to be higher than that in Wistar rats at 13 months of age (*p* < 0.02; Figure 3A). The density of mature neurons did not significantly change with age in either rat strain; however, a slight decrease of this parameter in Wistar rats and a slight increase of this parameter in OXYS rats caused elevation of mature neurons’ density in OXYS rats compared to Wistar rats at 13 months of age (*p* < 0.02; Figure 3A). The training in the MWM starting at 12 months of age increased the density of mature neurons only in Wistar rats (*p* < 0.04).

Therefore, we demonstrated that the training in the MWM did not affect the density of cells of the neuronal lineage in the DG with the exception of the density of mature neurons in control rats: This parameter increased in the DG of Wistar rats trained starting at 12 months of age.

### 2.5. Densities of Cells from the Astrocyte Cell Lineage in the DG of OXYS and Wistar Rats

The estimation of astrocyte progenitors’ density in the DG revealed that this parameter was affected by genotype (F_1.64_ = 11.4, *p* < 0.001) and training (F_1.64_ = 4.1, *p* < 0.05). We did not notice any age-related changes in astrocyte progenitors’ density; however, a slight decrease of the parameter from 2.5 to 13 months of age in Wistar rats and a slight increase of the parameter in OXYS rats caused higher density of astrocyte progenitors in the DG of 13-month-old OXYS rats (*p* < 0.004; Figure 4A). As for effects of MWM performance on astrocyte progenitors’ density, we registered a decrease of this parameter in 2.5-month-old Wistar rats (*p* < 0.03; Figure 4A).

The density of astrocytes was influenced by genotype (F_1.64_ = 43.7, *p* < 0.0001) and age (F_1.64_ = 70.5, *p* < 0.0001), and there was an interaction between genotype and age (F_1.64_ = 25.7, *p* < 0.0001). We observed a significant decline of astrocyte density from 2.5 to 13 months of age in both rat strains (*p* < 0.0001 for Wistar rats and *p* < 0.02 for OXYS rats); however, in OXYS rats, the decrease was less pronounced. A possible reason is lower density of astrocytes at 2.5 months of age in OXYS rats than in Wistar rats (*p* < 0.0001; Figure 4A). The training in the MWM did not significantly affect astrocyte density in Wistar rats with the exception of a slight increase in this parameter after the training started at 1.5 months of age. By contrast, we noted a 1.5-fold increase of astrocyte density in OXYS rats after the MWM training started at 12 months of age (*p* < 0.05; Figure 4A).

To summarize the obtained results, we can conclude that the MWM training that was started at 1.5 months of age drove accelerated maturation of astrocyte progenitors in the DG of Wistar rats but did not affect the astrocyte cell lineage in OXYS rats. We did not detect any impact of the MWM training on the astrocyte cell lineage in older Wistar rats; however, this was not the case for OXYS rats: We documented a slight decrease in astrocyte progenitors’ density and a significant increase in astrocyte density, meaning acceleration of their maturation.

### 2.6. The Influence of Spatial Learning on Amyloid-β Accumulation in the DG of OXYS Rats

At the final stage of the study, we analyzed APP and amyloid-β accumulation in the DG of OXYS and Wistar rats at age 13 months. We showed that APP levels (Figure 5A) were more than two-fold higher in OXYS rats than in Wistar rats (*p* < 0.02), whereas levels of amyloid-β (Figure 5B) did not significantly differ between the rat strains (*p* > 0.05). The training in the MWM starting at 12 months of age did not alter APP levels but decreased levels of amyloid-β in OXYS rats (*p* < 0.03).

## 3. Discussion

The MWM is a generally accepted instrument for studying spatial memory and learning, and there are several modifications of the standard protocol for different research goals [17,18]. Here we used a modification of the MWM, allowing us to examine not only spatial learning but also reversal learning in the rats. This protocol’s modification meant that rats were forced to swim in the MWM four times per day for 10 consequent days. Thus, we suppose that the experimental conditions included not only cognitive training but also to some extent physical training.

In the present work, we showed that at either 1.5 or 12 months of age, MWM performance was similar between OXYS and Wistar rats. This observation supports the results obtained earlier [19,20], which indicate that spatial learning in OXYS rats at 12 months of age does not significantly differ from that in Wistar rats. Here we found that the same is true for spatial reversal learning: Indeed, this parameter in OXYS rats did not differ from that in Wistar rats either at 1.5 months or at 12 months of age. Nonetheless, it should be pointed out that OXYS rats manifested altered reference memory at both studied ages. Taken together, our findings allow us to interpret the changes in neuro- and astrocytogenesis in the DG after the MWM training as a consequence of cognitive training in both rat strains.

Neurogenesis is a highly dynamic process and is modulated by multiple physiological stimuli and pathological states [21]. The duration of a neurogenic process from mitosis to maturation of newborn neurons—that is, mean excitability and morphology of mature neurons—takes 4 weeks [22]. At 4 weeks after mitosis, the majority of newborn neurons undergoes apoptosis [23], whereas the rescued newborn granule cells are recruited into circuits supporting spatial memory [24]. Thus, to estimate effects of cognitive training on different stages of neurogenesis in the DG, in the present work, we analyzed cell densities at 4 weeks after the beginning of MWM training and, as a consequence, 2.5 weeks after the end of the MWM training.

First, we analyzed changes in the DG after cognitive training that was started at 1.5 months of age. Four weeks after the beginning of training in the MWM—meaning “at 2.5 months of age”—we revealed activation of neurogenesis in the DG of OXYS rats: Indeed, we observed a shift from QNPs to ANPs in the DG of trained animals as compared to untrained animals. Thus, QNP and ANP densities in trained young-adult OXYS rats become the same as these parameters in the control age-matched Wistar rats. As for Wistar rats, we documented a decrease in ANP density after the training in the MWM, possibly indicating a decrease in neurogenesis as well as an increase of neuronal maturation in the DG. We did not note any changes in the density of cells of the neuronal lineage in either OXYS or Wistar rats after the training in the MWM. According to the data obtained by Encinas and colleagues [25], it takes ANPs 1 to 5 days to go through cell cycle events and become neuroblasts. Nevertheless, newborn cells undergo apoptotic selection but exit the cell cycle to become neuroblasts. Thus, our results may point to long-lasting effects of cognitive training on the activation of QNPs and on the start of cell cycle events in the DG; however, it seems like this activation of neurogenesis does not lead to a significant increase in the number of granule cells in the DG. At the same time, activation of neuronal progenitors in the DG as a response to the cognitive training apparently occurred later in OXYS rats than in Wistar rats. Another possible explanation of the obtained data may be a difference between OXYS and Wistar rats in cell cycle kinetics inside the DG in response to the MWM training. Kinetics of the cell cycle is a modifiable factor that depends on intrinsic as well as extrinsic stimuli. Indeed, Farioli–Vecchioli and colleagues have reported that cell cycle duration of neural stem and progenitor cells shortens in response to running [26]. Regarding the cells of the astrocyte lineage, the training in the MWM intensified maturation of astrocytes in the DG of Wistar rats. We can speculate that this enhancement of astrocyte maturation after cognitive training should occur in OXYS rats too, but we did not see it because of the possible delay of DG cells’ response to stimuli in OXYS rats.

The cognitive training that was started in the period of active accumulation of amyloid-β in OXYS rats, i.e., at 12 months of age, did not affect the density of neuronal stem cells and progenitors in the neurogenic niche; however, it influenced the density of mature cells. The density of mature neurons increased only in trained Wistar rats and reached the level of younger animals, possibly indicating increased neurogenesis in the DG right after the MWM training. Nonetheless, we did not detect an influence of the cognitive training on neuronal-cell density in OXYS rats at this age. Our data are in line with the findings made by Hüttenrauch and coworkers [9], who used environmental enrichment to implement physical and intellectual stimulation of mice. They showed strong dendritic branching of DCX-positive cells and increased density of granule neurons in the DG of transgenic AD mice housed in an enriched environment at 6 months of age but not at 12 months of age [9]. On the other hand, the density of astrocytes increased in OXYS rats after the training and reached the level of control untrained Wistar rats. Because astrocytes are in close contact with stem cells in neurogenic niches [27], and astrocytic support is crucial for appropriate functioning of neurons and synapses [28], the increased astrocyte density in the DG of OXYS rats after the cognitive training may be interpreted as a beneficial event intended to slow down the development of neurodegenerative processes.

It is well-known that in APP-transgenic mice, cognitive stimulation and voluntary exercise decrease amyloid deposition in the brain [29,30]. Here we estimated the effects of MWM training on APP and amyloid-β content of the DG in 12-month-old OXYS rats. Previously, we have reported increased APP levels in the hippocampus and frontal cortex of OXYS rats at 13 months of age [31]. Here, we confirmed these results and did not find effects of MWM training on the APP load in the DG of 13-month-old OXYS rats. As for amyloid-β content, we did not detect interstrain differences, but it was a limitation of the method used. Indeed, we estimated amyloid-β content using antibodies interacting with both amyloid-β_1-40_ and amyloid-β_1-42_ forms of amyloid-β. It is widely accepted that amyloid-β_1-42_ is much more toxic [32]. Previously, we have shown that the amyloid-β_1-42_/amyloid-β_1-40_ ratio is significantly increased in OXYS rats compared to Wistar rats at 12 months of age, whereas the total amount of amyloid-β does not differ so much. Moreover, amyloid-β–immunoreactive deposits have been detected in the cerebral cortex, hippocampus, thalamus, hypothalamus, and the brain stem of OXYS rats at age 15–18 months, and this was not the case for age-matched Wistar rats [15]. Here, we demonstrated that cognitive training led to a decrease of amyloid-β immunoreactivity in the hippocampal neurogenic niche during spontaneous age-related development of AD signs in OXYS rats.

Altogether, our results indicate that cognitive training—that is started at a young age prior to the development of neurodegeneration—causes signs of neuronal-progenitor activation in the hippocampal neurogenic niche. When started during active amyloid-β accumulation, the cognitive training not only decreases amyloid-β immunoreactivity in the neurogenic niche but also may increase its support by astrocytes. Consequently, according to our data, to activate neuronal progenitors in the DG, cognitive training should be started prior to first neurodegenerative changes, whereas cognitive training accompanying the accumulation of amyloid-β affects only astrocytic support of the neurogenic niche.

## 4. Materials and Methods

### 4.1. Ethical Approval

All the experimental procedures were in compliance with the Directive 2010/63/EU of the European Parliament and of the Council of 22 September 2010. The protocol of the animal study was approved by the Commission on Bioethics of the Institute of Cytology and Genetics, the Siberian Branch of the Russian Academy of Sciences (SB RAS, № 12000-496 of 2 April 1980), Novosibirsk, Russia. Every effort was made to minimize the number of animals used and their discomfort.

### 4.2. Animals

Male senescence-accelerated OXYS rats and age-matched male Wistar rats were obtained from the Breeding Experimental Animal Laboratory of the Institute of Cytology and Genetics, SB RAS ([RFMEFI61914X0005 and RFMEFI61914X0010]; Novosibirsk, Russia). The animals were kept under standard laboratory conditions (22 ± 2 °C, 60% relative humidity, and 12 light/12 h dark cycle) and had ad libitum access to standard rodent feed (PK-120-1, Laboratorsnab, Ltd., Moscow, Russia) and water. OXYS and Wistar rats at ages 1.5 and 12 months (*n* = 8 animals per strain and age) were trained in the MWM. Control groups consisted of age-matched OXYS and Wistar rats (*n* = 8 animals per strain and age) that were not trained in the MWM.

### 4.3. The MWM Test

This is a test of spatial learning for rodents that is based on distal cues to navigate from starting locations to a submerged escape platform [16,33]. The animals were trained in a circular open swimming arena (200 cm in diameter) located in a well-lit room with proximal visual cues on the walls of the maze and distal cues placed in the room. An escape platform (170 cm^2^) was submerged 2.0 cm below the surface of the water, which was maintained at 22 ± 2 °C and mixed with powdered milk to obscure the platform. Two principal axes of the maze were designated dividing the arena into four equal quadrants. The platform was positioned in the middle of the first quadrant throughout the 5-day training period. Trials (five consecutive days with four trials per day) were conducted with the same hidden platform location using a semirandom set of start locations; one trial each day was initiated from each of the four positions. Each trial either lasted for 70 s or ended when a rat reached the submerged platform, thus escaping water. To assess spatial reversal learning, the platform was relocated to the third quadrant, and another set of four trials per day for 5 additional days was administered. At the end of the learning, one probe trial was given 24 h after the last acquisition day, that is, on the eleventh day. Reference memory was quantified using a preference for the platform area when the platform was absent in the probe trial. Latencies of finding the submerged escape platform throughout the training period and the time spent in the third quadrant in the probe trial were measured by means of a computerized video system.

### 4.4. Tissue Preparation

Four weeks after the beginning of learning, rats were euthanized by CO_2_ asphyxiation and decapitation; the brains were carefully excised, and hemispheres were separated. For an immunohistochemical assay, the hemispheres were immediately fixed in 4% paraformaldehyde in phosphate-buffered saline (PBS) at room temperature (RT) for 48 h, followed by cryoprotection in 30% sucrose in PBS at 4 °C for 48 h. Then, the brains were frozen and stored at −70 °C until further processing.

### 4.5. Immunohistochemistry

Brain sagittal sections (20 μm thick) of OXYS and Wistar rats from the training and control groups (*n* = 3 to 6 per group, strain, and age) were prepared on a Microm HM-505 N cryostat (Microm, Walldorf, Germany) at −20 °C and transferred onto polysine-glass slides (Menzel-Glaser, Braunschweig, Germany). After serial washes with PBS, the slices were incubated at RT for 15 min in PBS-plus (PBS with 0.1% of Triton X-100) and for 1 h in 3% bovine serum albumin (BSA; cat. # A3294, Sigma-Aldrich, St. Louis, MO, USA) in PBS to permeabilize the tissues and to block nonspecific binding sites, and then were incubated overnight with primary antibodies at 4 °C. The primary antibodies were all diluted 1:250 with 3% BSA in PBS; these were antibodies to nestin, vimentin, GFAP, DCX, Fox-3 (NeuN), amyloid precursor protein (APP), and amyloid-β (MOAB2) (cat. ## ab6142, ab24525, ab7260, ab54739, ab177487, ab15272 (Abcam, Cambridge, MA, USA), and MABN254 (Merck, Darmstadt, Germany), respectively). After several washes with PBS, the slices were incubated with secondary antibodies conjugated with Alexa Fluor 488, 568, or 555 (cat. ## ab150073, ab175472, and ab150170, respectively, Abcam, Cambridge, MA, USA) in PBS (1:250) for 1 h at RT and next were washed in PBS. The slices were coverslipped with the Fluoroshield mounting medium containing 4′,6-diamidino-2-phenylindole (DAPI; cat. # ab104139, Abcam, Cambridge, MA, USA). Negative controls were processed in an identical manner except that a primary antibody was not included. The nestin, vimentin, GFAP, DCX, NeuN, APP, and MOAB2 signals were counted under a microscope with a 40× objective lens (Axioskop 2 plus, Zeiss, Oberkochen, Germany). The microscopy was conducted at the Multi-Access Center for Microscopy of Biological Objects (Institute of Cytology and Genetics, SB RAS, Novosibirsk, Russia). Identification of cell types was carried out according to protein markers described by Encinas and colleagues [24]. To evaluate the density of quiescent (nestin-positive and vimentin-positive) and amplifying (nestin-positive) neural progenitors, neuroblasts (DCX-positive), immature (DCX-positive and NeuN-positive) and mature (NeuN-positive) neurons, as well as astrocyte progenitors (vimentin-positive and GFAP-positive) and astrocytes (GFAP-positive), the total number of counted cells was divided by the area of the DG and was presented as the number of cells per 10,000 μm^2^. To assess the impact of spatial learning starting at 12 months of age on amyloid-β accumulation, we calculated the mean intensity of immunofluorescence of APP and amyloid-β in the DG of OXYS and Wistar rats.

### 4.6. Statistics

The data were subjected to three-way analysis of variance (ANOVA) in the Statistica 8.0 software (StatSoft, Tulsa, OK, USA). The genotype, age, and MWM training were chosen as independent variables. The Newman–Keuls post hoc test was applied to significant main effects and interactions in order to assess the differences between some sets of means. The *t*-test for dependent samples was performed for dependent-pair comparison. The data were presented as mean ± standard error of the mean (SEM). The differences were considered statistically significant at *p* < 0.05.

## Figures and Tables

**Figure 1 ijms-21-06986-f001:**
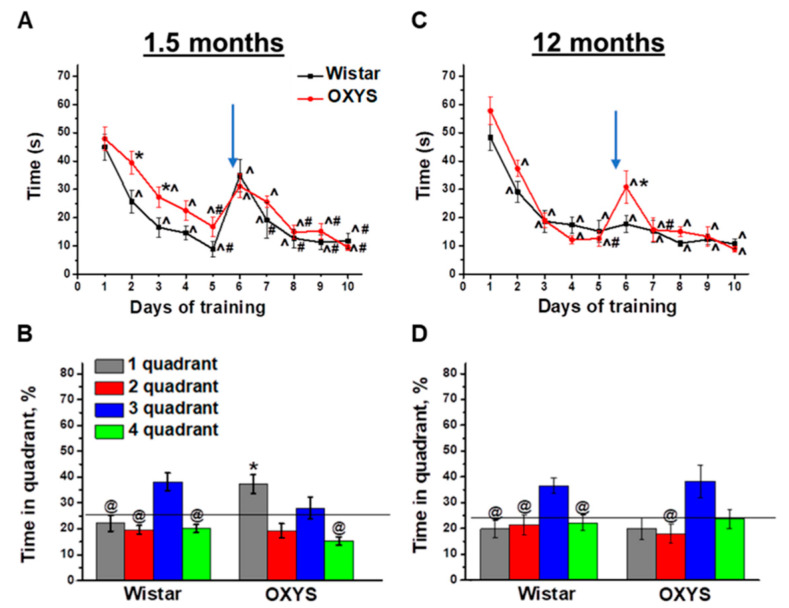
Morris ware maze (MWM) performance of OXYS and Wistar rats at 1.5 and 12 months of age. Time required for finding a submerged platform on subsequent training days for OXYS and Wistar rats at 1.5 months of age (**A**) and 12 months of age (**C**). In the probe trial, 1.5-month-old Wistar rats preferred the target third quadrant, whereas OXYS rats remembered the previous location of the platform in the first quadrant (**B**). At 12 months of age, Wistar rats preferred the target third quadrant to all other quadrants; this was not the case for OXYS rats (**D**). The data are presented as mean ± SEM, *n* = 8. * *p* < 0.05 for differences between the strains; ^ *p* < 0.05 for a comparison with the first day of testing; ^#^
*p* < 0.05 for a comparison with the sixth day of testing; ^@^
*p* < 0.05 for a comparison with time in the third quadrant. The arrow indicates the move of the platform from the first to third quadrant. The black line (**B**,**D**) indicates the average time the rats should spend in each quadrant without training.

**Figure 2 ijms-21-06986-f002:**
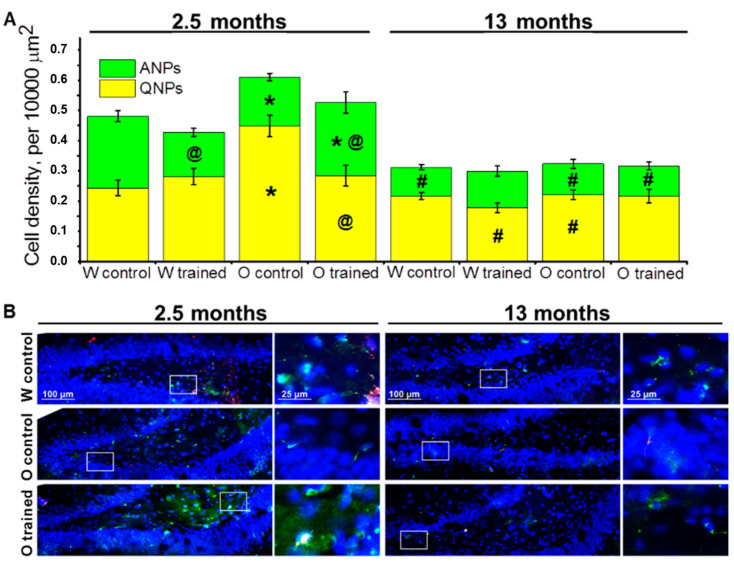
Neuronal progenitors’ density in the dentate gyrus (DG) of OXYS and Wistar rats. (**A**) The density of quiescent neural progenitors (QNPs) was higher in 2.5-month-old OXYS rats than in Wistar rats, and training in the MWM decreased this parameter in OXYS rats to the level of Wistar rats. The density of ANPs at 2.5 months of age was lower in OXYS rats compared to Wistar rats, and the training in the MWM lowered the parameter in young Wistar rats but increased it in young OXYS rats. The data are presented as mean ± SEM, *n* = 3 to 6. * *p* < 0.05 for differences between the strains; ^#^
*p* < 0.05 for a comparison with the previous age; ^@^
*p* < 0.05 for effects of the MWM training. W: Wistar rats; O: OXYS rats. (**B**) Photomicrographs of the DG of control Wistar and OXYS rats, as well as trained OXYS rats at ages 2.5 and 13 months are shown as representative images of immunohistochemical staining with antibodies against nestin (green) and vimentin (red). DAPI (blue) highlights cell nuclei.

**Figure 3 ijms-21-06986-f003:**
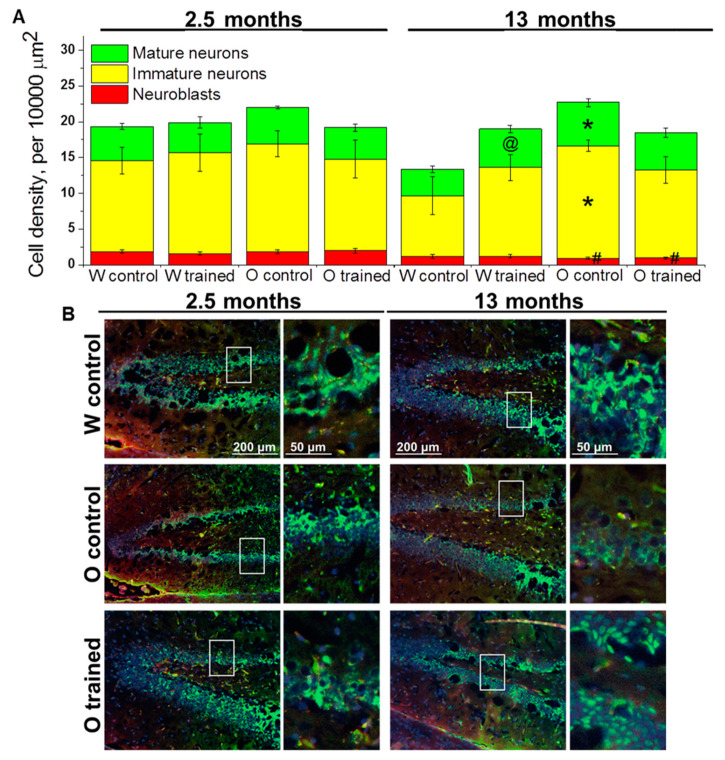
Density of cells of the neuronal lineage in the DG of OXYS and Wistar rats. (**A**) Genotype and MWM training did not influence the density of cells of the neuronal lineage at 2.5 months of age. At 13 months of age, the density of immature and mature neurons was higher in OXYS rats than in Wistar rats; training in MWM led to an increase of mature neurons’ density in Wistar rats. The data are presented as mean ± SEM, *n* = 3 to 6. * *p* < 0.05 for differences between the strains; ^#^
*p* < 0.05 for a comparison with the previous age; ^@^
*p* < 0.05 for effects of the MWM training. W: Wistar rats; O: OXYS rats. (**B**) Photomicrographs of the DG of control Wistar and OXYS rats, as well as trained OXYS rats at ages 2.5 and 13 months are shown as representative images of immunohistochemical staining with antibodies against NeuN (green) and doublecortin (DCX; red). DAPI (blue) highlights cell nuclei.

**Figure 4 ijms-21-06986-f004:**
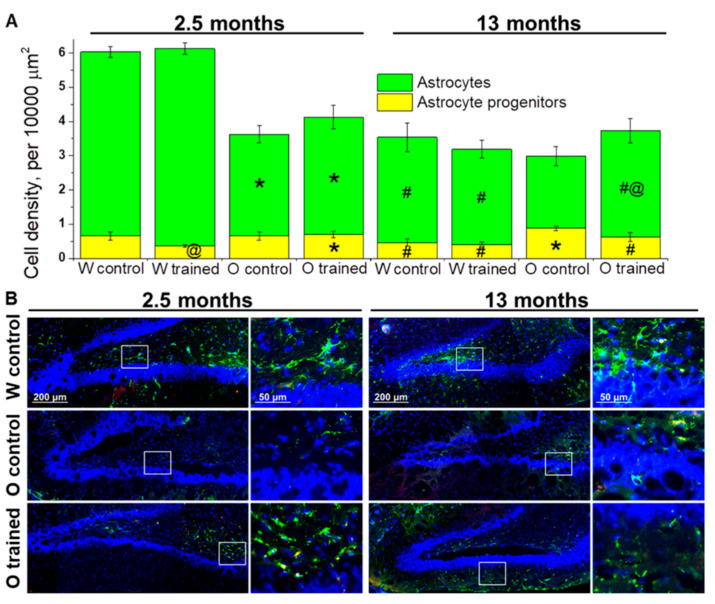
Density of cells of the astrocyte lineage in the DG of OXYS and Wistar rats. (**A**) At age 2.5 months, the density of astrocytes was lower in OXYS rats than in Wistar rats; training in the MWM resulted in faster maturation of astrocyte progenitors in Wistar rats. At 13 months of age, the density of astrocyte progenitors was higher in OXYS rats compared to Wistar rats, and the MWM training increased astrocytes’ density in OXYS rats. The data are presented as mean ± SEM, *n* = 3 to 6. * *p* < 0.05 for differences between the strains; ^#^
*p* < 0.05 for a comparison with the previous age; ^@^
*p* < 0.05 for effects of the MWM training. W: Wistar rats; O: OXYS rats. (**B**) Photomicrographs of the DG of control Wistar and OXYS rats, as well as trained OXYS rats at ages 2.5 and 13 months are shown as representative images of immunohistochemical staining with antibodies against glial fibrillary acid protein (GFAP; green) and vimentin (red). DAPI (blue) highlights cell nuclei.

**Figure 5 ijms-21-06986-f005:**
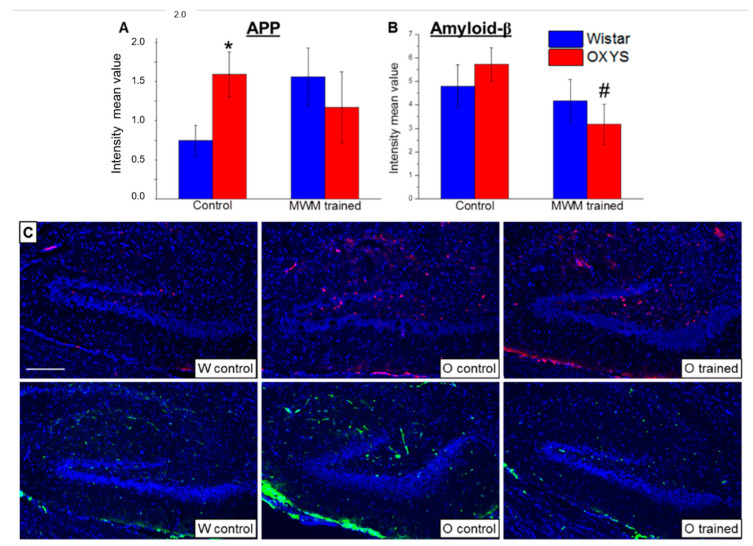
Mean immunofluorescence intensity of APP and amyloid-β in the DG of OXYS and Wistar rats at age 13 months. The load of APP (**A**) was higher in OXYS rats than in Wistar rats. MWM training did not affect APP content but decreased levels of amyloid-β (**B**) in OXYS rats. The data are presented as mean ± SEM, *n* = 6. * *p* < 0.05 for differences between the strains; ^#^
*p* < 0.05 for effects of the MWM training. W: Wistar rats; O: OXYS rats. (**C**) Photomicrographs of the DG of untrained Wistar and OXYS rats and a trained OXYS rat are shown as representative images of immunohistochemical staining with antibodies against APP (red) and amyloid-β (green). DAPI (blue) highlights cell nuclei. The scale bar (**C**) is 200 μm.

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
