# Peer review of "Cognitive Training as a Potential Activator of Hippocampal Neurogenesis in the Rat Model of Sporadic Alzheimer’s Disease"

_ijms, 2020, doi:10.3390/ijms21196986_

Round 1
Reviewer 1 Report
In this study, Burnyasheva and collegues examined the effects of cognitive
training on neurogenesis in the dentate gyrus of presymptomatic and symptomatic rats affected by AD.
Their conclusion is that that the cognitive training activated neuronal progenitors only when initiated in young presymptomatic animals. In fact, when started during active amyloid-β accumulation, the cognitive training not only reduced the levels of amyloid-β.
however, as main concern, I would to underlyine that the data shown are too preliminar to substantiate the conclusion.
Many other experiments should be done on the expression of specific markers of neurogenesis together with those of amyloid-β accumulation.
In this respect it is necessary to show some specific experiments (and data) on amyloid-β accumulation in these animals during the disease progression.
Furthermore, the innunolocalization images should be depicted at different magnification together with the appropriate images of AD group at the same age.
Colletively, I advise authors to better charcterize their AD model and to show appropriate experiments to corroborate their conclusion.
.
Author Response
Thanks a lot for the important comments, we have tried to address all of them.
however, as main concern, I would to underlyine that the data shown are too preliminar to substantiate the conclusion.
We have changed the conclusion, so that it is more reasonable and consistent with the obtained results (lines 300–310).
Many other experiments should be done on the expression of specific markers of neurogenesis together with those of amyloid-β accumulation.
Previously, we have shown changes in the expression of genes associated with adult neurogenesis (according to a mammalian-adult-neurogenesis Gene Ontology database) in the hippocampus of OXYS rats using RNA-Seq data. We have added a discussion of these results in the Introduction section (lines 50–53 and 55–57).
In this respect it is necessary to show some specific experiments (and data) on amyloid-β accumulation in these animals during the disease progression.
Sorry for not mentioning this before. We have previously published data about amyloid-β accumulation in OXYS rats; accordingly, we have inserted this information into the Discussion section (lines 288–292 and 297–299).
Furthermore, the innunolocalization images should be depicted at different magnification together with the appropriate images of AD group at the same age.
As requested, we have added to Figures 2–4 images with different magnification as well as control and trained AD groups at different ages.
Colletively, I advise authors to better charcterize their AD model and to show appropriate experiments to corroborate their conclusion.
We have inserted previously published data about amyloid pathology and changes of neurogenesis in OXYS rats into the Introduction and Discussion sections (lines 50–53, 55–57, 288–292 and 297–299).
Reviewer 2 Report
This study makes a good effort in trying to explore the effects of cognitive training in the Morris water maze on neurogenesis in the dentate gyrus in presymptomatic (young 16 rats) and symptomatic (adult rats) periods of development of AD signs.
The manuscript has a clear and concise structure. The information is provided in a comprehensible manner and reflects an enormous amount of work.
Only a change should be made: the Material and Methods section should be moved before the Results section.
I thank the authors for the interesting study.
Author Response
Only a change should be made: the Material and Methods section should be moved before the Results section.
Thank you for the positive evaluation of this manuscript. Unfortunately, we cannot agree with the placement of this section. In the Instructions for Authors for IJMS, there is a requirement that the Materials and Methods section should be located after the Discussion section.
Reviewer 3 Report
The manuscript titled as ‘Cognitive training as a potential activator of hippocampal neurogenesis in the rat model of sporadic Alzheimer’s disease’ by Burnyasheva et al. examined effects of cognitive training in the Morris water maze on neurogenesis in the dentate gyrus in presymptomatic (young rats) and symptomatic (adult rats) periods of development of AD signs. The authors found that cognitive training did not affect neuronal-lineage cells’ density in both rat strains either at the young or adult age but activated neuronal progenitors in young rats and increased astrocyte density and downregulated amyloid-β in adult OXYS rats. The authors conclude that to activate adult neurogenesis, cognitive training should be started before first neurodegenerative changes, whereas cognitive training accompanying amyloid-β accumulation affects only astrocytic support. This work should be of wide interests to most researchers on neuroscience and molecular medicine etc.
This manuscript has sufficient novel and findings and the method described is highly practical. I recommend that the manuscript be accepted with some revisions. The following points need to be addressed:
- The data analysis is mainly from the comparison between the groups of OXYS and Wistar rats, or between the groups of rats at age 2.5 months and 13 months, there is not n values indicated in figure legends, although this has been mentioned in the method section.
- There are some statistical analysis and comparison between the groups of rats at age 2.5 months and 13 months, however, the data is presented in A panel and B panel of figures, is that easy for readers to understand?
- There is not representative images of immunohistochemical staining for Figure 5.
Author Response
Thank you for the important comments! We have tried to address all of them.
1. The data analysis is mainly from the comparison between the groups of OXYS andWistar rats, or between the groups of rats at age 2.5 months and 13 months, there is not n values indicated in figure legends, although this has been mentioned in the method section.
Sorry about this oversight. We have added the n values into figure legends (lines 107, 150, 178, 205, 497).
2. There are some statistical analysis and comparison between the groups of rats at age 2.5 months and 13 months, however, the data is presented in A panel and B panel of figures, is that easy for readers to understand?
We have combined graphs for the 2.5 and 13 months into one panel for Figures 2–4.
3. There is not representative images of immunohistochemical staining for Figure 5.
As suggested, we have added images of immunohistochemical staining in Figure 5.
Round 2
Reviewer 1 Report
With this new revision, the manuscript has been improved and now it is suitable for publication